# Microsolvation of a Proton by Ar Atoms: Structures and Energetics of Ar_n_H^+^ Clusters

**DOI:** 10.3390/molecules29174084

**Published:** 2024-08-28

**Authors:** María Judit Montes de Oca-Estévez, Rita Prosmiti

**Affiliations:** 1Institute of Fundamental Physics (IFF-CSIC), CSIC, Serrano 123, 28006 Madrid, Spain; juditmontesdeoca@iff.csic.es; 2Atelgraphics S.L., Mota de Cuervo 42, 28043 Madrid, Spain

**Keywords:** ab initio electronic structure calculations, molecular interactions, machine learning potentials, noble gas proton-bound clusters, microsolvation structures

## Abstract

We present a computational investigation on the structural arrangements and energetic stabilities of small-size protonated argon clusters, 
${\mathrm{Ar}}_{n}$
$H^{+}$. Using high-level ab initio electronic structure computations, we determined that the linear symmetric triatomic 
${\mathrm{ArH}}^{+}$Ar ion serves as the molecular core for all larger clusters studied. Through harmonic normal-mode analysis for clusters containing up to seven argon atoms, we observed that the proton-shared vibration shifts to lower frequencies, consistent with measurements in gas-phase IRPD and solid Ar-matrix isolation experiments. We explored the sum-of-potentials approach by employing kernel-based machine-learning potential models trained on CCSD(T)-F12 data. These models included expansions of up to two-body, three-body, and four-body terms to represent the underlying interactions as the number of Ar atoms increases. Our results indicate that the four-body contributions are crucial for accurately describing the potential surfaces in clusters with 
n> 3. Using these potential models and an evolutionary programming method, we analyzed the structural stability of clusters with up to 24 Ar atoms. The most energetically favored 
${\mathrm{Ar}}_{n}$
$H^{+}$ structures were identified for magic size clusters at *n* = 7, 13, and 19, corresponding to the formation of Ar-pentagon rings perpendicular to the 
${\mathrm{ArH}}^{+}$Ar core ion axis. The sequential formation of such regular shell structures is compared to ion yield data from high-resolution mass spectrometry measurements. Our results demonstrate the effectiveness of the developed sum-of-potentials model in describing trends in the nature of bonding during the single proton microsolvation by Ar atoms, encouraging further quantum nuclear studies.

## 1. Introduction

Small-size noble gas hydride cations have aroused recent interest [1,2,3,4,5,6,7,8], as they could help enhance knowledge about the stability and behavior of the new systems discovered in the interstellar medium (ISM), such as the 
${\mathrm{HeH}}^{+}$ [9] and 
${\mathrm{ArH}}^{+}$ [10]. Consequently, they offer a strong starting point [11,12,13,14,15,16,17,18] for investigating the formation of higher-order complexes, when solvated by more than one noble gas (Ng) atom. Solvent species may interact strongly with the dissolved ions, forming a solvation shell around the charged center [19,20], with the number of solvent atoms needed for shell closure depending on the balance of the underlying interactions.

From a computational point of view, such 
$[{\mathrm{Ng}}_{n}$
${H]}^{+}$ aggregates are extremely challenging systems, as quantum anharmonic treatments are required for studying their fully coupled dynamics and spectroscopy, and recent studies have been reported for the *n* = 2 and 3 cases [2,21,22,23,24,25]. They are known as proton-bound complexes (PBCs), which are characterized by being exceptionally bright molecules in the infrared (IR) region, with a high intensity of the stretching vibrational mode of the shared central proton [26,27,28]. This fact makes them really easy targets to detect experimentally; thus, in recent years, there has been a revival of interest in studying these noble gas hydride cations by advanced computational quantum chemistry and experimental laboratory techniques, aiming to understand the chemical nature of these compounds and their evolution in the ISM [1].

From the experimental point of view, the detection of the 
${\mathrm{Ar}}_{2}$
$H^{+}$ PBC was first reported in solid Ar/Kr matrix isolation spectroscopy experiments [29,30], and later on through electron bombardment matrix isolation of Ng/methanol mixtures [31], characterized by absorption bands at 905 and 903.8 
${\mathrm{cm}}^{-1}$, respectively, while recently, Duncan and coworkers [8] have recorded the first gas phase infrared photodissociation (IRPD) spectra for the 
${\mathrm{Ar}}_{n}$
$H^{+}$ (*n* = 3–7) clusters. These spectra show a series of strong bands, with those of the lowest frequency shifted gradually from 989 to 965 
${\mathrm{cm}}^{-1}$ as the size of the cluster increases from *n* = 3 to 7, respectively. Additionally, in this latter study [8], the geometry optimizations at Møller–Plesset perturbation, MP2/AVTZ, level of theory, for all clusters up to *n* = 7. They have revealed that the [
${\mathrm{ArH}}^{+}$Ar] core structure remained practically invariant as more Ar atoms were added to the complex, in line with recent results from a coupled cluster with single, double, and perturbative triple excitations, CCSD(T)/AVTZ, and calculations on the structure and vibrational spectra of 
${\mathrm{Ar}}_{3}$
$H^{+}$ [2]. More recently, high-resolution mass spectroscopy measurements have been reported on protonated and pristine He and Ar clusters of up to 50 noble gas atoms [3,4,5], indicating significant differences between the heaviest Ar and He protonated species, identifying magic sizes for the 
${\mathrm{Ar}}_{n}$
$H^{+}$ for *n* = 7, 13, and 19.

In this context, the present work aims to characterize small finite-size 
${\mathrm{Ar}}_{n}$
$H^{+}$ clusters, up to 24 Ar atoms. As such clusters consist of Ar atoms weakly bound to the covalent centro-symmetric Ar-
$H^{+}$-Ar core, we have first investigated the interactions of the linear [
${\mathrm{ArH}}^{+}$Ar] trimer with an additional Ar atom, performing high-level ab initio coupled cluster explicit correlated F12, CCSD(T)-F12/AVQZ, calculations. Both the configuration and energy datasets of these calculations have proven fundamental in training a machine learning (ML) potential energy surface (PES) model by employing a 2D reproducing kernel Hilbert space (RKHS) scheme [32,33]. In turn, we explored various simplified sum-of-potentials approaches, such as those including expansion up to two-body (
2B), three-body (
3B), and four-body (
4B) terms, to represent the Ar-doped proton groups as the number of Ar atoms increases. Such analytical potential models were validated against ab initio MP2, CCSD(T), and CCSD(T)-F12 data using AVQZ and AV6Z basis sets [34,35,36,37]. The structural characteristics and energetics of the 
${\mathrm{Ar}}_{n}$
$H^{+}$ clusters were investigated by potential optimizations through an evolutionary programming (EP) algorithm [38,39]. Comparisons with available data in the literature were discussed, contributing to confirm certain effects during the proton microsolvation by Ar atoms.

## 2. Computational Details, Results and Discussion

### 2.1. Binding and Structuring in 
${\mathrm{Ar}}_{n}$
$H^{+}$ Clusters from Electronic Structure Calculations

As a starting point in exploring how the Ar atoms are arranged around the proton in the higher-order 
${\mathrm{Ar}}_{n}$
$H^{+}$ clusters, geometry optimizations and frequency analysis were performed at CCSD(T)/AVQ/6Z, CCSD(T)-F12/AVQZ, and MP2/AVQZ levels of theory, depending on the size of the cluster. All ab initio electronic structure calculations were performed with MOLPRO 2022 program [40], while the DENEB software package [41] was employed to generate and organize all input and output data files, respectively.

Various initial guess configurations, some of them given in previous studies on such systems [24,42], were considered in order to identify the most stable conformer for each cluster. Given that the number of electrons involved increases substantially when a new Ar atom is added, we have considered performing the optimization calculations at the MP2 level of theory with the AVQZ basis set.

Figure 1 and Appendix A present our results on optimized structures derived from the MP2/AVQZ and CCSD(T)/AV6Z or CCSD(T)-F12/AVQZ calculations performed on the 
${\mathrm{Ar}}_{n}$
$H^{+}$ clusters, and the binding energies with respect the dissociation of one Ar atom from each cluster, 
${\mathrm{Ar}}_{n}$
$H^{+}$→ 
${\mathrm{Ar}}_{n-1}$
$H^{+}$ + Ar. By examining the corresponding configurations, one can observe a common core structure in each 
${\mathrm{Ar}}_{n}$
$H^{+}$ cluster, consisting of a central linear [
${\mathrm{ArH}}^{+}$Ar], that remains almost unchanged as additional Ar atoms are added to the cluster. One can note that the first two Ar atoms form a relatively strong bond of about 5500 
${\mathrm{cm}}^{-1}$, compared to the rest of the argons that bond more weakly, with energies of 600–900 
${\mathrm{cm}}^{-1}$ (see Appendix A). Similarly, the bond distances of the core are notably shorter than in the rest of the unions (1.5 Å vs. 3.2/4.4 Å), as seen in Figure 1. Ar atoms outside the core align perpendicularly to the molecular axis of [
${\mathrm{ArH}}^{+}$Ar] until the complex reaches a size of *n* = 7, where the first solvation shell is completed, forming a pentagonal dipyramid structure. For larger clusters with *n* = 8 and 9, the Ar atoms are accommodated at significantly larger distances, around 4.4 Å from the core. From such initial analysis of the structural characteristics of these cations, it is clear that the binding in the proton-bound dimer is strong, while additional Ar atoms form weakly bound clusters. This indicates that the interactions between this cationic core and a single Ar atom could represent the most dominant contributions in the description of the PESs of higher-order 
${\mathrm{Ar}}_{n}$
$H^{+}$ clusters. With this in mind, the next step entails examining the intermolecular interactions in the 
${\mathrm{Ar}}_{3}$
$H^{+}$.

### 2.2. Building Up a ML PES for the 
${\mathrm{Ar}}_{3}$
$H^{+}$

The generation of high-quality data that represent the entire configurational space is a fundamental process in the construction of any PES. Here, we used Jacobi coordinates (*r*, *R*, 
θ) to define the 
${\mathrm{Ar}}_{3}$
$H^{+}$ system (see inset plot in the upper panel of Figure 2), where *r* is the vector along the Ar–
$H^{+}$–Ar distance, *R* is the vector along the distance between the Ar atom and the center of mass of the [
${\mathrm{ArH}}^{+}$Ar] core, and 
θ is the angle between the (
r,R) vectors. Since the structure of the [
${\mathrm{ArH}}^{+}$Ar] core remains nearly unchanged as the size of the 
${\mathrm{Ar}}_{n}$
$H^{+}$ clusters increases, we have decided to keep the *r* coordinate fixed to the equilibrium bond length of the 
${\mathrm{ArH}}^{+}$Ar at 
$r_{e}$ = 3.0112 Å. In this way, the configuration space is sampled using the *R* and 
θ descriptors in the interval of *R* = 2.0–8.0 Å and 
θ = 0–
$90^{\circ }$ (see Figure 2).

Next, the performance of the electronic structure method employed is also an important issue in the PES construction. Thus, regarding the choice of the level of theory, and taking into account the increased demand of the computational resources, together with the outcome of previous benchmark studies of such complexes [24], a way to maintain the quality of the data is through the use of the CCSD(T)-F12 methodology employing the AVQZ basis set. As has been demonstrated in ref. [24], the CCSD(T)-F12 energies closely reproduce CCSD(T)/CBS[56] data. On this basis, the CCSD(T)-F12/AVQZ method was used for the present calculations, with the interaction energy defined as 
$\Delta E=E_{Ar_{3}H^{+}}-E_{Ar_{2}H^{+}}-E_{Ar}-\delta_{BSSE}$, with 
$E_{Ar_{3}H^{+}}$, 
$E_{Ar_{2}H^{+}}$, and 
$E_{Ar}$ being the total energies of the 
${\mathrm{Ar}}_{3}$
$H^{+}$ complex and 
${\mathrm{Ar}}_{2}$
$H^{+}$ and Ar fragments, respectively, while 
$\delta_{BSSE}$ includes the basis set superposition error (BSSE) correction [43].

Among the ML-approaches, we have chosen to employ the RKHS method, which is a reasonable option due to its successful results with low dimensional systems and small training sets. Briefly, if 
V(x) is the potential energy function with 
$V_{i}$ corresponding to known potential energies at different molecular configurations 
$x_{i}$, then using the representer theorem for a general functional relationship, the 
V(x) can be optimally approximated as a linear combination of suitable functions, 
$V\left(x\right)\approx V_{i}=\sum_{i=1}^{N}C_{i}K\left(x,x_{i}\right)$, where 
$C_{i}$ are coefficients and 
$K(x,x^{\prime })$ is a kernel function. The coefficient 
$C_{i}$ can be determined for all input 
y = 
$\left[V_{1},V_{2},\cdots ,V_{N}\right]$ and 
$x_{i}$ in the data set by solving the linear equation 
$C=K^{-1}y$, with 
$C=[C_{1},\cdots ,C_{N}]$ being the vector coefficients, and 
K being the 
N×N reproducing kernel matrix. The explicit expression of multidimensional kernel functions 
$K(x,x^{\prime })$ can be written as a direct product, 
$K\left(x,x^{\prime }\right)=\prod_{d=1-D}k\left(x,x^{\prime }\right)$, with D being the dimension of the multidimensional *K* kernel, and *k* are the one-dimensional (1D) kernels. Several 1D kernel functions for different purposes are available in the literature [44], and in this study, we have employed the 
$k_{1}^{n,m}$ and 
$k_{2}$ 1D reproducing kernel functions for the distance-like (*R*) and angle-like (
θ) variables [32], respectively, which combine kernels with physically motivated descriptors, like those determining the smoothness of the kernel function and its correct asymptotic behavior. The reduced coordinates are 
x=R and 
y=cosθ; 
$N_{R}$ and 
$N_{\theta }$ are the number of ab initio calculated points in each *R* and 
θ coordinate, with
(1)
$$k_{1}^{n,m}\left(x,x^{\prime }\right)=n^{2}x_{>}^{-(m+1)}B{\left(m+1,n\right)}_{2}F_{1}\left(-n+1,m+1;n+m+1;\frac{x_{<}}{x_{>}}\right)$$
where 
$x_{>}$ and 
$x_{<}$ are the larger and smaller of the *x* and 
$x^{\prime }$, respectively, and
(2)
$$k_{2}\left(y,y^{\prime }\right)=\sum_{ll}\frac{\left(2ll+1\right)}{2}P_{ll}\left(y\right)P_{ll}\left(y^{\prime }\right)$$

The *n* and *m* superscripts refer to the order of smoothness of the function and its asymptotic behavior at large distances, with 
n=2, and *m* = 3, as the 
$R^{-4}$ accounts for the leading dispersion interaction between the Ar atom and 
${\mathrm{ArH}}^{+}$Ar molecular cation. *B* is the beta function, 
F12 is the Gauss hyper-geometric function [32], and 
$P_{ll}$ are the Legendre polynomials, with 
ll = 2*k* and *k* = 0–6. In turn, the potential form is given by 
$V\left(R,\theta ;r_{e}\right)=\sum_{i=1}^{N_{R}}\sum_{j=1}^{N_{\theta }},C_{ij}k_{1}^{n,m}\left(x_{i},x\right)k_{2}\left(y_{j},y\right)$, where the 
$C_{ij}$ coefficients are obtained by solving the linear equations, with V(
$R_{i}$, 
$\theta_{k}$;*r*) being the ab initio CCSD(T)-F12/AVQZ energy at each (
$R_{i}$, 
$\theta_{j}$, 
$r_{e}$) grid point.

We have generated a total of 2679 ab initio CCSD(T)-F12/AVQZ datasets, with 1159 of them used in the training step, and 1520 in the testing step. The training points were equally distributed across a 61-point grid in the internuclear distance *R*, ranging from 2.0 to 8.0 Å, and a 19-point grid was used in the angle 
θ coordinate between 0 and 
$90^{\circ }$. We have employed a hold-out cross-validation scheme to select the best performing RKHS PES among six models trained on 200, 380, 400, 760, 610, and 1159 points, respectively. These points were selected with variable number of 20, 40, or 61 grid points in the *R* coordinate, and 10 or 19 in the 
θ coordinate, covering the whole radial and angular intervals. Figure 3 depicts the histograms of these training data sets and, as one can see, the energy distributions look quite similar, although by increasing the number of angular grid points permits a broader sampling, covering the whole energy range.

Next, in the left panel of Figure 4, we display histograms of the energy distribution of the 1520 CCSD(T)-F12/AVQZ configurations used for the PES evaluation during the testing process. In the right panel of Figure 4, we show the root-mean-squared error (RMSE) values as a function of the training dataset size. One can observe the systematic improvement of the RMSE of the RKHS ML-PES models as the size of the training dataset increases. The RMSE values were computed by averaging over the total number of 1520 testing data and over 506 randomly chosen points, as well as over 1341 and 445 randomly chosen test data with energies below the dissociation threshold. The quality of the six RKHS ML-PES models as a function of the size of the training data with 200, 380, 400, 760, 610, and 1159 points is demonstrated in the right panel of Figure 4. As the size of the dataset increases, the corresponding RMSE value decreases. We also observed a large sensitivity in the RMSE values with respect to the sampling in the 
θ coordinate. In particular, we found that the RKHS ML-PES models trained on the 200 (20 × 10), 400 (40 × 10), and 610 (61 × 10) datasets show much higher RMSE values compared to those obtained from the 380 (20 × 19), 760 (40 × 19), and 1159 (61 × 19) ones. Additionally, we found that the calculated RMSE values for all six RKHS ML-PES trained models with respect to the total 1520 data, 1341 datasets with energies below the dissociation threshold, and the two corresponding 506/445-randomly chosen testing sets exhibit the same behavior.

Correlation plots are shown in Figure 5, demonstrating the performance of the chosen RKHS ML-PES model, which corresponds to that trained to 1159 CCSD(T)-F12, data against the energies of both training (see left panel) and testing (see right panel) data in both the attractive and repulsive regions of the potential. The RMSE values are also displayed as a function of energy ranges (see inset plot in the right panel of Figure 5). One can see that the RMSE values outside the training zone are 24 
${\mathrm{cm}}^{-1}$ up to dissociation energies over 1341 configurations (see also the left panel of Figure 4), and a MAE lower than 0.07% over 179 configurations and up to energies of 600 
${\mathrm{cm}}^{-1}$ above dissociation.

Once we have constructed the 2D RKHS ML-potential for the 
${\mathrm{Ar}}_{3}$
$H^{+}$ cation, we then analyzed the specific characteristics of its topology. To facilitate this analysis and ensure the smoothness of the PES, we present 1D curves (see Figure 2) and a 2D contour plot (see Figure 6).

One can clearly observe the presence of two minima on the 
${\mathrm{Ar}}_{3}$
$H^{+}$ PES. The global minimum corresponds to the T-shaped configuration with a well-depth of 38,888.67 
${\mathrm{cm}}^{-1}$ for the RKHS ML-PES and CCSD(T)-F12/AVQZ data, located at *R* = 3.20 Å (0.06 Å larger than that predicted by the MP2/AVQZ optimization calculation; see Figure 1). The negligible difference between the ab initio data and the present RKHS ML-PES suggests that the RKHS model describes accurately and smoothly this region of the potential surface. The second minimum lies around 250 
${\mathrm{cm}}^{-1}$ above the global one, and corresponds to the linear geometry of the complex, Ar-Ar-Ar-
$H^{+}$, at an energy of −38,634.80 
${\mathrm{cm}}^{-1}$ with R = 4.90 Å. Again, there is an excellent agreement of these PES values compared to those from the ab initio CCSD(T)-F12 data at −38,655 
${\mathrm{cm}}^{-1}$ and 4.93 Å, respectively.

### 2.3. Modeling the Interaction in Higher-Order 
${\mathrm{Ar}}_{n}$
$H^{+}$ Clusters

In the ongoing effort to characterize and understand the nature of interactions within higher-order size 
${\mathrm{Ar}}_{n}$
$H^{+}$ clusters, we have employed the many-body expansion formalism [45], which is a fundamental approach for calculating the total energy (
$E_{n}$) of systems consisting of *n*-bodies or monomers as
(3)
$$E_{n}=\sum_{n}E_{i}^{\left(1B\right)}+\sum_{n}E_{ij}^{\left(2B\right)}+\sum_{n}E_{ijk}^{\left(3B\right)}+\ldots +E^{\left(nB\right)}$$
where each term in the sum series reflects the contribution of specific interactions between *n*-bodies, such as the pairwise two-body interaction, 
$E_{ij}^{\left(2B\right)}$, and the higher nonadditive three-body, 
$E_{ijk}^{\left(3B\right)}$, up to *n*-body, 
$E^{\left(nB\right)}$, terms. Through such decomposition, investigation on complex systems are significantly simplified, and could provide a solid foundation for the analysis of physical and chemical phenomena. In this context, we will explore the capability of three analytical schemes based on the sum of two (
2B)-, three (
3B)-, and four-body (
4B) interactions to represent and model the underlying interactions in 
${\mathrm{Ar}}_{n}$
$H^{+}$ clusters. The corresponding expressions are given by
(4)
$$V_{2B}\left(\vec{R_{i}}\right)=\sum_{i=1-N}V_{{\left[ArH\right]}_{i}^{+}}\left(\vec{R_{i}}\right)+\sum_{i,l=1-N,l>i}V_{ArAr}\left(\vec{R_{il}}\right)$$
(5)
$$V_{3B}\left(\vec{R_{i}},r\right)=\sum_{i=1-N}V_{{\left[Ar_{2}H\right]}_{i}^{+}}\left(\vec{R_{i}},r\right)+\sum_{i,l=1-N,l>i}V_{ArAr}\left(\vec{R_{il}}\right)$$
(6)
$$V_{4B}\left(\vec{R_{i}},r_{e}\right)=\sum_{i=1-N}V_{{\left[Ar_{3}H\right]}_{i}^{+}}\left(\vec{R_{i}},r_{e}\right)+\sum_{i,l=1-N,l>i}V_{ArAr}\left(\vec{R_{il}}\right)$$
where the vectors 
$R_{i}$ connect the proton with the *i*-th Ar atom, with polar and azimuthal angles 
$\theta_{i}$ and 
$\phi_{i}$, respectively, while 
r and 
$r_{e}$ are the vectors along the 
${\mathrm{ArH}}^{+}$ and 
${\mathrm{AHAr}}^{+}$ in Equations (Equation 5) and (Equation 6), respectively (see left and right panels of Figure 7, respectively), and 
$R_{il}$ vectors connect the *i*-th and *l*-th Ar atoms in each 
${\mathrm{Ar}}_{n}$
$H^{+}$ cluster (see Figure 7). The corresponding 
$V_{{\mathrm{ArH}}^{+}}\left(R\right)$ and 
$V_{{\mathrm{Ar}}_{2}H^{+}}\left(R,r\right)$ terms are the RKHS ML-PESs of 
${\mathrm{ArH}}^{+}$ and 
${\mathrm{Ar}}_{2}$
$H^{+}$ from refs. [24] and [46], respectively, while the 
$V_{{\mathrm{Ar}}_{3}H^{+}}\left(R,r_{e}\right)$ corresponds to the present 
${\mathrm{Ar}}_{3}$
$H^{+}$ RKHS ML-PESs. The 
$V_{\mathrm{ArAr}}$ is the potential for 
${\mathrm{Ar}}_{2}$ from ref. [47].

### 2.4. Assessing the Sum-of-Potentials Models

In order to validate the three different sum-of-potential approaches adapted here (
$V_{2B}$, 
$V_{3B}$, and 
$V_{4B}$), we have also carried out additional CCSD(T)-F12/AVQZ calculations for the pentatomic 
${\mathrm{Ar}}_{4}$
$H^{+}$ cluster.

Figure 8 displays the potential energy curves obtained from the different sum-of-potentials analytical expressions (see Equations (Equation 4)–(Equation 6)), along with the corresponding ab initio computed interaction energies at CCSD(T)/CBS[56] for the *n* = 2 cluster along the 
$R_{1}$ coordinate, and CCSD(T)-F12/AVQZ for the *n* = 3 and 4 clusters along the 
$R_{2}$ and 
$R_{3}/R_{4}$ coordinates, respectively. As illustrated in the upper panels of Figure 8, we explored the cases with the 
${\mathrm{ArH}}^{+}$*r* value fixed at its equlibrium 1.5056 Å distance, and thus the 
${\mathrm{ArH}}^{+}$Ar core *r* value is 3.0112 Å. In the *n* = 4 case, we considered selected configurations with fixed *r* and 
$R_{3}$ bondlengths at 3.0112 Å and 3.12 Å, respectively, or 
$R_{3}$ = 
$R_{4}$ with Ar atoms in T-shaped geometries (as shown in the inset plots in the lower panels).

One can see that for all the cases studied, the form of additive atom–atom interactions (
$V_{2B}$) predicts potential energies that differ by around 5000 
${\mathrm{cm}}^{-1}$ in the well-depth region, as compared to the corresponding ab initio CCSD(T)/CBS[56] data. This fact reveals that three-body interactions will be important in the construction of 
${\mathrm{Ar}}_{2}$
$H^{+}$ PESs, as confirmed by the results obtained when one employs the 
$V_{3B}$ sum-of-potential form. In this case, the differences in that region are significantly reduced, counting 200 
${\mathrm{cm}}^{-1}$ for 
${\mathrm{Ar}}_{3}$
$H^{+}$ and 
${\mathrm{Ar}}_{4}$
$H^{+}$, as seen in the upper-right and lower-left panels of Figure 8, respectively. On the other hand, in the second configuration of 
${\mathrm{Ar}}_{4}$
$H^{+}$ (see the lower-right panel in Figure 8), the difference increases up to 400 
${\mathrm{cm}}^{-1}$. Regarding the 
$V_{4B}$ sum-of-potential model, we observe that deviations between interaction energies are not greater than 20 
${\mathrm{cm}}^{-1}$ in both 
${\mathrm{Ar}}_{4}$
$H^{+}$ configurations. For that reason, we choose the 
$V_{4B}$ model, based on the sum of the CCSD(T)-F12/AVQZ 
$V_{Ar_{3}H^{+}}$ interactions, for all 
${\mathrm{Ar}}_{n}$
$H^{+}$ clusters with 
n> 3 in the present study.

### 2.5. Searching for 
${\mathrm{Ar}}_{n}$
$H^{+}$ Potential Energy Minima and Microsolvated Structures

Once the performance of the sum-of-potential approach was assessed, we proceed with searching for minimum energy structures on the 
$V_{4B}$ potentials for each 
${\mathrm{Ar}}_{n}$
$H^{+}$ cluster, and compare them with those obtained from the MP2/AVQZ calculations. Geometry optimization calculations on high-dimensional potential surfaces is a rather challenging task as the size of the molecular system increases. Thus, we employed an evolutionary programming (EP) algorithm [38] to localized the lowest energy structures for each 
${\mathrm{Ar}}_{n}$
$H^{+}$ cluster on the corresponding 
$V_{4B}$ PES. Such EP algorithms have demonstrated their accuracy and efficiency in solving numerical optimization problems in multi-dimensional space [38,39,48].

Briefly, the process begins by generating an initial population of 
M=100 individuals for each 
${\mathrm{Ar}}_{n}$
$H^{+}$ cluster under consideration, with *n* varying from 1 to 24. Each individual is characterized by a pair of real vectors 
$\left(\chi_{i},\eta_{i}\right)$ for 
i=1 to *M*, containing the Cartesian coordinates 
$\left(\chi_{i}\right)$ of all cluster atoms and their standard deviation 
$\left(\eta_{i}\right)$ for Gaussian mutation (strategy parameter), governing the evolution of population dispersion over time. The initial coordinates 
$\chi_{i}$, with 
$\eta_{i}=1$, are randomly selected in the interval 
(0,Δ), where 
Δ is a displacement factor enhancing resolution. Each parent set 
$\left(\chi_{i},\eta_{i}\right)$ evolves to generate a new population through mutation [38], while for each individual in the joint parent–child group 
(2M individuals), *q* (tournament size equals 100) opponents are randomly chosen from the 
2M−1 individuals for comparison. The individual with the lower potential energy in each encounter emerges victorious. The top *M* individuals then serve as parents for the next generation, and this process continues iteratively. Convergence is achieved when the potential energy difference between two consecutive generations falls below a threshold value of 
$10^{-3}$

${\mathrm{cm}}^{-1}$. Thus, by employing the EP optimization procedure for the potential surface of Equation (Equation 6), we were able to calculate the corresponding minima for the 
${\mathrm{Ar}}_{n}$
$H^{+}$ clusters. The obtained results are summarized in Figure 9 and Appendix A in comparison with the MP2/AVQZ and CCSD(T)-F12 values (see also in Figure 1), as well as with previously data available in the literature [8]. Both set of data consistently demonstrate good agreement, with differences not exceeding 30 
${\mathrm{cm}}^{-1}$ in most cases, corresponding to errors of around 5%. In Figure 9, we also included zero-point energy (ZPE) corrections, 
$E_{\mathrm{ZPE}}$, in the MP2/AVQZ energy values, by treating each cluster in the framework of the harmonic approximation. The corresponding ZPE values are obtained from normal-mode frequency calculations at each cluster’s optimized structure, and are found to significantly affect the energetics of the clusters, although one should consider such values as an upper limit, taking into account the contribution of anharmonities.

In this context, Figure 10 presents comparisons for the fundamental asymmetric Ar–
$H^{+}$ stretching (or proton-shared) vibration frequencies between the harmonic normal mode values at MP2/AVQZ level of theory and the vibrational bands observed in the experimentally recorded gas phase IRPD spectra [8] as the size of the 
${\mathrm{Ar}}_{n}$
$H^{+}$ clusters increases from *n* = 2 to 7, as well as with that from solid Ar-matrix isolation measurements [29]. At first glance, one can clearly observe that the proton-shared harmonic frequency is shifted to lower values as the cluster size increases, in accord with the behavior observed in the gas phase IRPD spectra. However, the frequencies from the harmonic normal-mode analysis are found to be higher by 75–82 
${\mathrm{cm}}^{-1}$ than the IRPD reported values for *n* = 3 to 7 clusters [8], and also by 95 
${\mathrm{cm}}^{-1}$ compared to the anharmonic value reported by quantum calculations in the *n* = 2 case [24]. We should further notice that the gradual shift of the proton stretch vibration to lower frequencies as the cluster size increases reaches a limited value near to 1044 or 965 
${\mathrm{cm}}^{-1}$ from the harmonic approach and IRPD experiment, respectively. Such finding indicates the first shell closing at *n* = 7 in the gas phase clusters, although by comparing these frequency values with that observed in solid Ar matrix experiments at 905 
${\mathrm{cm}}^{-1}$ seems that additional shells and different local arrangements might also have an influence.

Finally, the stability of each cluster size was investigated concerning its nearest neighbors. In Figure 11, we illustrate the single-atom evaporative energies, defined as 
$E_{n-1}$–
$E_{n}$, and the average energy per added Ar atom, 
$E_{n}/n$ (see inset plot). These quantities were determined through both EP optimizations using the 
$V_{4B}$ potential, and MP2/AVQZ optimization calculations, and compared with previously reported dissociation energies for loosing a single Ar atom for the cationic cluster [5]. The dependence of the cluster size on the evaporation energy shows pronounced steps at *n* = 4 and 7, indicating the completion of the first solvation shell or the formation of a particularly stable structure. It is observed that results from both EP and ab initio calculations exhibit similar behavior starting from the *n* = 4 cluster. By examining the obtained 
$E_{n}/n$ values, it is interesting to note that only the first two Ar atoms are independent and strongly bound to the 
$H^{+}$, yielding about −32,440 and −19,120 
${\mathrm{cm}}^{-1}$
$E_{n}/n$ energies, respectively. For all sequential Ar atoms, this value is reduced, indicating a progressive importance of the Ar–Ar interaction. We found that the average energies per Ar atom approach a plateau as *n* increases, around −5000 
${\mathrm{cm}}^{-1}$ at *n* = 9. This behavior implies that the addition of new Ar atoms does not significantly alter the energetics of the larger clusters.

Further, the computed evaporation energies are also compared (see in Figure 11 with experimental values of 
${\mathrm{Ar}}_{n}$
$H^{+}$ ion yields up to 50 Ar atoms, as measured recently by high-resolution mass spectroscopy experiments [3]. By now examining the extracted distributions from ion yield measurements and the evaporation energies for the 
${\mathrm{Ar}}_{n}$
$H^{+}$ clusters, one can see in Figure 11 the standout features corresponding to magic size clusters at *n* = 7, 13 and 19 in both data, as well as weaker peaks at *n* = 23, 26, 29 and 32 in the ion yield data. In Figure 9 and Appendix A, we depict the computed optimal structures corresponding to these size clusters, and as one can clearly observe, the first solvation shell is built up by a central pentagon ring in the equatorial plane around the proton perpendicular to the 
${\mathrm{ArH}}^{+}$Ar core ion axis, while the 13-atom cluster features the formation of a second pentagon ring at the one end of the 
${\mathrm{ArH}}^{+}$Ar axis together with an additional Ar atom along it. The symmetrical accommodation of the remaining Ar atoms on the other side of the central ring at distances of around 4.45 Å corresponds to the optimal 
${\mathrm{Ar}}_{19}$
$H^{+}$ structure. On can also observe (see in Appendix A) that for *n* = 20 starts a new shell, and as the cluster size increases the Ar atoms are accommodated accordingly.

## 3. Summary and Conclusions

We have investigated in detail the energetic and structural characteristics of 
${\mathrm{Ar}}_{n}$
$H^{+}$ clusters (with *n* up to 24). As a starting point, optimizations were carried out on small-size cations using the MP2/AVQZ level of theory, and a common cationic [
${\mathrm{ArH}}^{+}$Ar] core was observed in all of them. Such a finding has been verified by a normal-mode vibrational analysis showing red shifts in the asymmetric Ar-
$H^{+}$ stretch frequency by increasing the cluster size from *n* = 2 to 7, in agreement with the gas phase IRPD experimental measurements. In order to address the interaction of the 
${\mathrm{ArH}}^{+}$Ar core with the addition of new Ar atoms, a two-dimensional (2D) ML-PES based on the RKHS method and trained on CCSD(T)-F12/AVQZ datasets has been constructed for the 
${\mathrm{Ar}}_{3}$
$H^{+}$ complex. Next, analytical sum-of-potentials representations were generated through expansions up to two-body (
$V_{2B}$), three-body (
$V_{3B}$), and four-body (
$V_{4B}$) interactions. These potentials were validated by direct comparisons with ab initio energies, noting that the 
$V_{4B}$ form accurately predicts the CCSD(T)-F12 reference data for 
n> 3 clusters. Thus, such analytical potential expression has been then considered to describe the intermolecular interactions in the higher-order 
${\mathrm{Ar}}_{n}$
$H^{+}$ clusters up to 24 Ar atoms.

The 
$V_{4B}$ potential models employed in optimization EP calculations, and in comparison with the MP2/AVQZ results, have been provide information on the growth of the 
${\mathrm{Ar}}_{n}$
$H^{+}$ clusters. According to the minimum energy structures obtained, the solvation of the proton occurs symmetrically around its two ends for small size clusters of up to 7 Ar atoms, forming a dipyramidal structure around the 
$H^{+}$. These results agree with the ab initio electronic structure calculations, suggesting that the present 
$V_{4B}$ PESs allow extracting qualitative and quantitative knowledge about the energy and structural arrangements during the microsolvation processes of the 
$H^{+}$ cation in Ar atoms. In the future, it would be interesting to investigate the importance of quantum anharmonic, as well as temperature effects on the clusters structure and stability, and whether, for larger clusters, the optimized configurations remain symmetric or whether, on the contrary, the dominant influence of Ar-Ar interactions leads to more compact structures in subsequent solvation shells. Our studies could contribute to a comprehensive understanding of the energetic and structural characteristics of 
${\mathrm{Ar}}_{n}$
$H^{+}$ clusters, as well as shedding light on interactions that take place in such systems during the proton’s microsolvation by Ar atoms, and assign patterns observed experimentally.

## Figures and Tables

**Figure 1 molecules-29-04084-f001:**
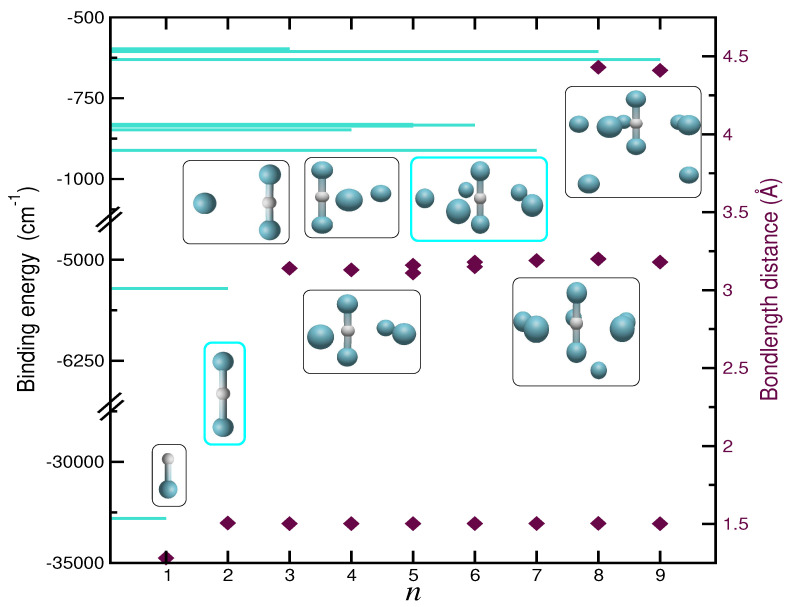
Binding energies and bondlengths of the optimized structures (see inset plots) of the 
${\mathrm{Ar}}_{n}$
$H^{+}$ complexes (with *n* = 1–9) from CCSD(T)/AV6Z (*n* = 1–2) and MP2/AVQZ (*n* = 3–9) calculations.

**Figure 2 molecules-29-04084-f002:**
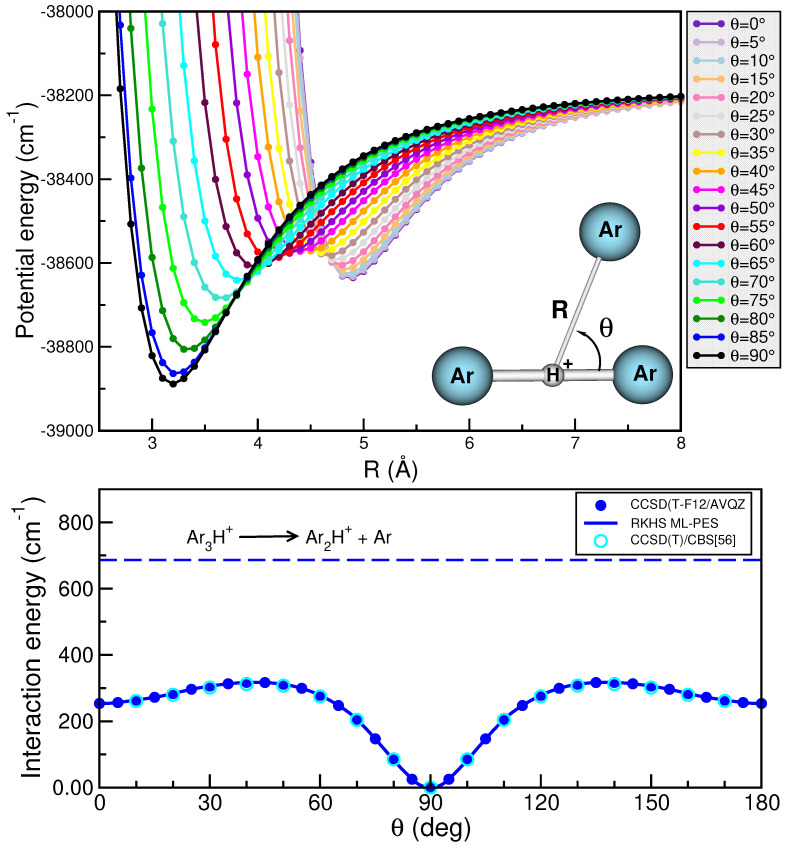
Interaction CCSD(T)-F12/AVQZ energies (see circle symbols) as a function of *R* distance at the indicated 
θ values (see **upper** panel) and minimum energy path along both *R* and 
θ coordinates as a function of 
θ angles (see **lower** panel) for the 
${\mathrm{Ar}}_{3}$
$H^{+}$ cation (see inset plot). The corresponding RKHS ML-PES curves are also shown as solid lines, while the dissociation energy is plotted by long-dashed line.

**Figure 3 molecules-29-04084-f003:**
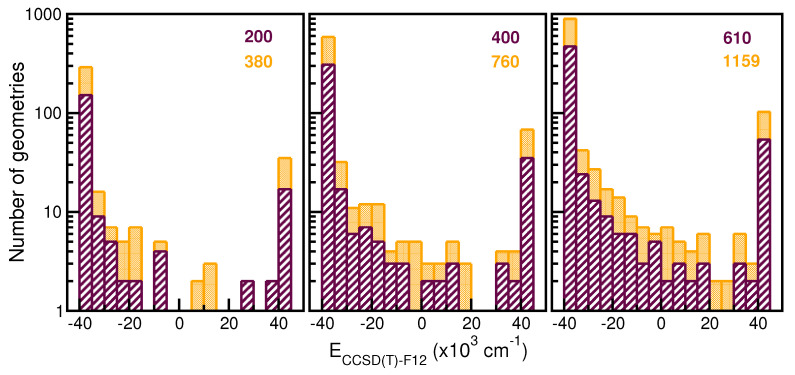
Histograms of training datasets employed in the hold-out cross-validation scheme for assessing the RKHS ML-PES models. The number of total configurations considered is given in the top of each panel. The left, middle and right side panels correspond to increasing numbers of grid points in *R*, 20, 40 and 61, respectively, while the orange and maroon colors show datasets with 10 and 19 grid points in 
θ.

**Figure 4 molecules-29-04084-f004:**
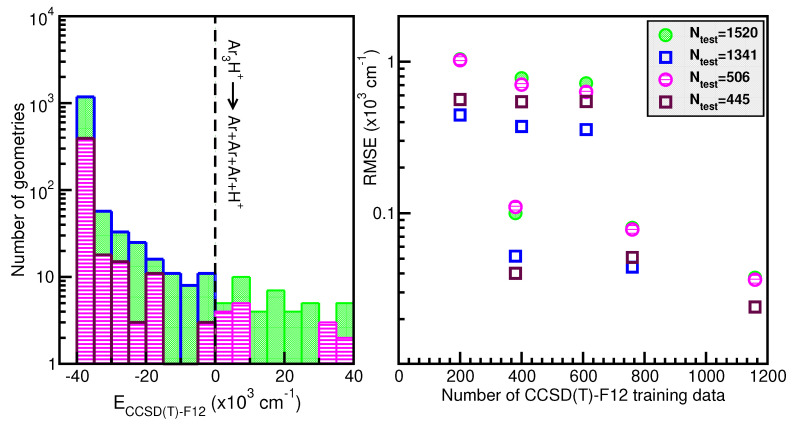
Histograms of the total and randomly generated testing datasets (see text) employed in hold-out cross-validation scheme for the RKHS ML-PES (**left** panel). The corresponding RMSE values of the RKHS ML-PES models as a function of the training set size (**right** panel). The black dashed line indicates the dissociation threshold.

**Figure 5 molecules-29-04084-f005:**
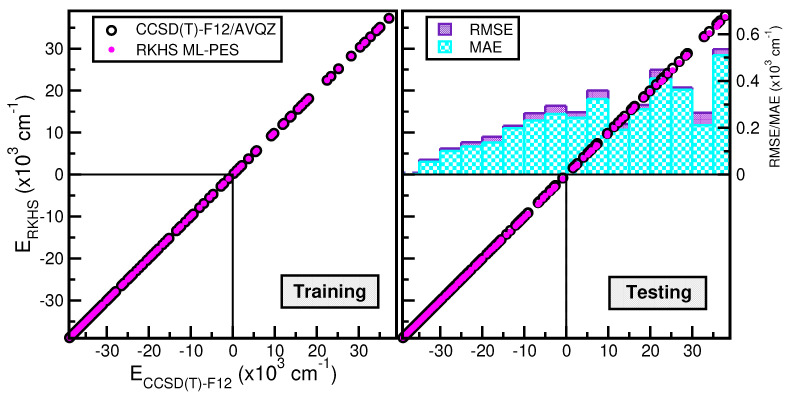
Correlation plots of the RKHS ML-PES model against the reference CCSD(T)-F12/AVQZ energies for both training (**left** panel) and testing (**right** panel) data. The corresponding average RMSE and MAE values along energy are also plotted.

**Figure 6 molecules-29-04084-f006:**
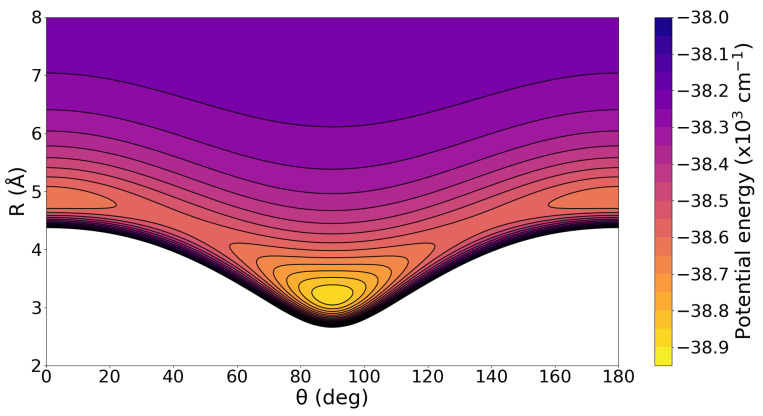
2D contour plot of the RKHS ML-PES for 
${\mathrm{Ar}}_{3}$
$H^{+}$ complex in the (
θ,R)-plane. The equipotential curves are at energies of −39,000 to −38,000 
${\mathrm{cm}}^{-1}$ in intervals of 100 
${\mathrm{cm}}^{-1}$.

**Figure 7 molecules-29-04084-f007:**
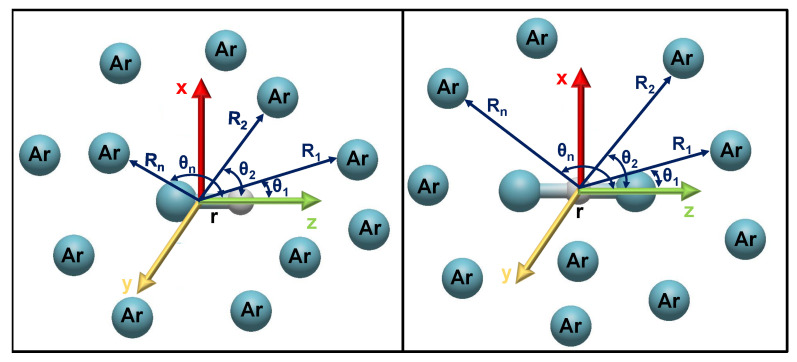
Coordinate system used for the 
${\mathrm{Ar}}_{n}$
$H^{+}$ clusters in the sum-of-three-body (**left** panel) and sum-of-four-body (**right** panel) potential approaches.

**Figure 8 molecules-29-04084-f008:**
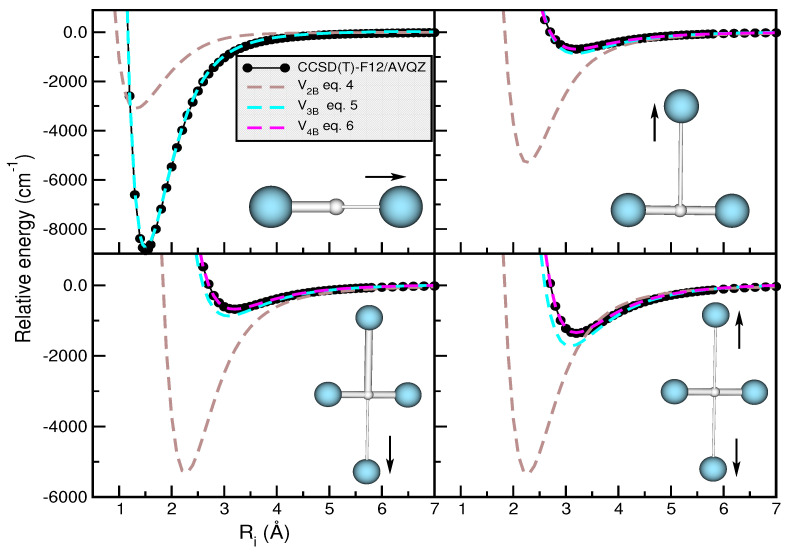
Potential curves obtained for the 
${\mathrm{Ar}}_{n}$
$H^{+}$ (with *n* = 2, 3 and 4) clusters using the sum-of-potential approaches of Equations (Equation 4)–(Equation 6) (see color dashed lines), together with the calculated CCSD(T)/CBS[56] (for *n* = 2) and CCSD(T)-F12/AVQZ (for *n* = 3 and 4) interaction energies (black circles) as a function of indicated 
$R_{i}$ coordinates.

**Figure 9 molecules-29-04084-f009:**
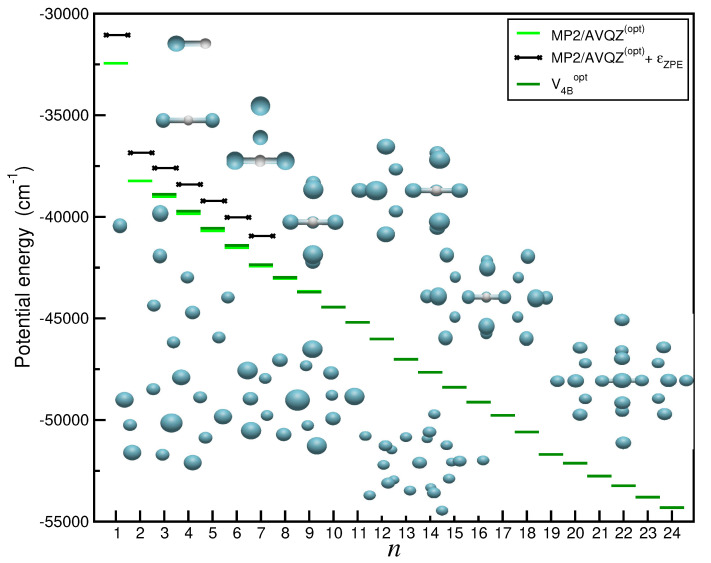
Schematic representation of selected optimal low-lying structures and their energetics obtained using the 
$V_{4B}$ approach of Equation (Equation 6) and the GA algorithm for the 
${\mathrm{Ar}}_{n}$
$H^{+}$ clusters. Energy values from the MP2/AVQZ optimizations and their ZPE corrections, 
$E_{\mathrm{ZPE}}$, given by the corresponding harmonic approximation, are also shown.

**Figure 10 molecules-29-04084-f010:**
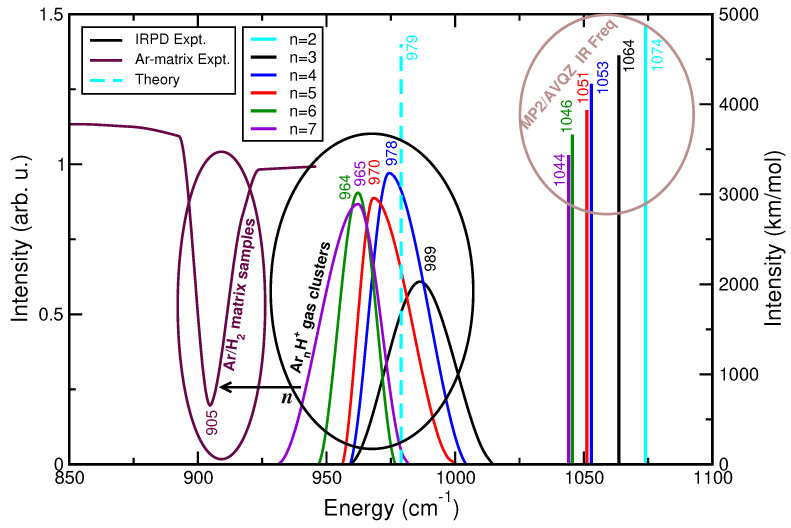
Computed harmonic frequencies for the indicated 
${\mathrm{Ar}}_{n}$
$H^{+}$ clusters from MP2/AVQZ calculations in comparison with data from gas phase IRPD [8] and solid Ar-matrix isolation measurements [29]. The anharmonic fundamental frequency for 
${\mathrm{Ar}}_{2}$
$H^{+}$ from ref. [24] is also displayed.

**Figure 11 molecules-29-04084-f011:**
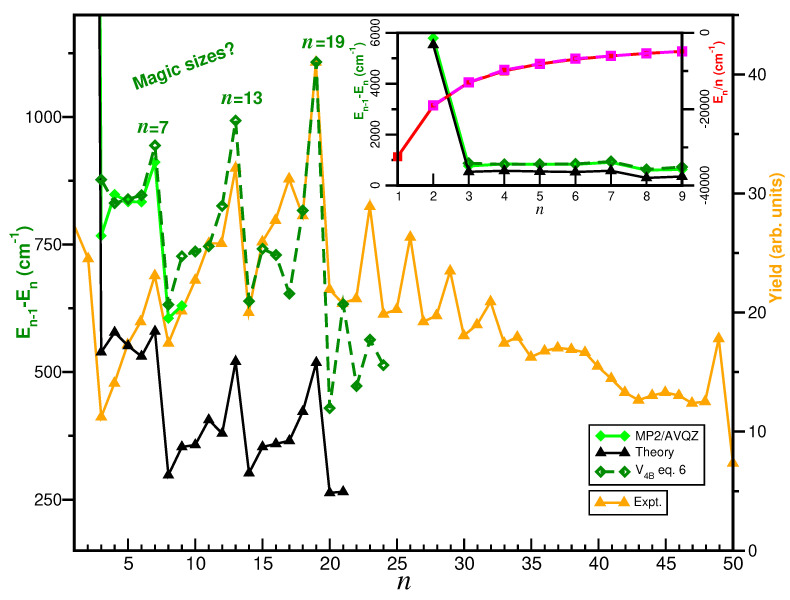
Computed single-atom evaporative energies (green color) of 
${\mathrm{Ar}}_{n}$
$H^{+}$ clusters, their average energy per Ar atom (red color) from the 
$V_{4B}$ potential EP optimizations (dashed lines) and MP2/AVQZ calculations (solid lines) as a function of *n*. Previously reported theoretical dissociation energies [5] and experimental ion yield values [3] are also displayed for comparison reasons.

## Data Availability

The data supporting reported results are available from the corresponding author upon request.

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
