# Peer review of "Microsolvation of a Proton by Ar Atoms: Structures and Energetics of ArnH+ Clusters"

_molecules, 2024, doi:10.3390/molecules29174084_

Round 1

Reviewer 1 Report

Comments and Suggestions for Authors

In this manuscript, the authors study the structural and energetic properties of
protonated argon clusters, combining high-level abinitio electronic calculations  
with kernel-based machine learning potential models. They find magic sizes clusters
at N=7, 13 and 19 corresponding to ArH+Ar core ion surrounded by argon atoms
distributed in pentagonal rings.

I think that this work is very interesting with new physical insights
and the results are clear and reasonable. I advice publishing this work,
but only after several revisions are made (see below):

a) In page 2, I recommend that the authors clearly define what the binding energy is, because the results presented in Figure 1 are not clear.

b) Authors should use the same notation to identify clusters, as in some cases it is used
in square brackets, [Ar_nH]+, and in others not, Ar_nH+. This can create confusion as in the first case it appears that the charge is distributed on the aggregate, but in the second case it is placed on the hydrogen atom.

c) In page 4, in the lower panel of Figure 2, the authors do not
indicate the value of R=R_min employed in the calculation.

d) In page 10, in the figure 9, the authors include the energy values from the
MP2/AVQZ optimizations in comparison with the lowest energy structures obtained with
the classical potential V_4b. Do the structures optimized with MP2/AVQZ match those
obtained with V_4b? Are they global minimum structures?

Author Response

Referee's comments  and Authors' reply

In this manuscript, the authors study the structural and energetic properties of protonated argon clusters, combining high-level abinitio electronic calculations with kernel-based machine learning potential models. They find magic sizes clusters at N=7, 13 and 19 corresponding to ArH+Ar core ion surrounded by argon atoms distributed in pentagonal rings.

I think that this work is very interesting with new physical insights and the results are clear and reasonable. I advice publishing this work, but only after several revisions are made (see below):

Authors' reply: We thank the reviewer for the supportive discussion on our work, and we have addressed all points raised in the revised ms.

a) In page 2, I recommend that the authors clearly define what the binding energy is, because the results presented in Figure 1 are not clear.

Authors' reply: We have now provide the binding energy definition on page 3, lines 91-92.

b) Authors should use the same notation to identify clusters, as in some cases it is used in square brackets, [Ar_nH]+, and in others not, Ar_nH+. This can create confusion as in the first case it appears that the charge is distributed on the aggregate, but in the second case it is placed on the hydrogen atom.

Authors' reply: We have standardized the notation throughout the revised ms.

c) In page 4, in the lower panel of Figure 2, the authors do not indicate the value of R=R_min employed in the calculation.

Authors' reply: We have now specified in the caption of Figure 2 in the revised ms, indicating that the MEP is along both the R and θ coordinates.

d) In page 10, in the figure 9, the authors include the energy values from the MP2/AVQZ optimizations in comparison with the lowest energy structures obtained with the classical potential V_4b. Do the structures optimized with MP2/AVQZ match those obtained with V_4b? Are they global minimum structures?

Authors' reply: In Figure 9 we present the optimized structures obtained using the V_4B potential model, as mentioned in both the figure caption and the text, which corerspond to the global minima structures of each cluster size. The corresponding optimized stuctures from the MP2/AVQZ calculations for clusters up to n=9 are shown in Figure 1. A comparison reveals that they follow the same growth pattern with similar bond lengths. We have added a reference to Figure 1 on page 11, line 265 in the revised ms.

Reviewer 2 Report

Comments and Suggestions for Authors

In this work, the authors investigate the structural arrangements and energetic stabilities of small protonated argon clusters using first-principle initio computations. It shows four-body contributions are crucial for accurately describing potential surfaces in clusters with more than three Ar atoms. Analyzing structural stability using these models and evolutionary programming, the study identifies energetically favored ArnH+ structures at magic sizes n=7, 13, and 19, forming Ar-pentagon rings perpendicular to the ArH+Ar core. These findings align with ion yield data from mass spectrometry and demonstrate the model's effectiveness.

This work can be interesting for the computational chemistry and the industrial community. I would like to ask the authors to consider the minor comments below.

1. page 2, line 88

“Figure 1 and Table S1 (see supplementary material) present our results on optimized structures and binding energies derived from the MP2/AVQZ and CCSD(T)/AV6Z or CCSD(T)-F12/AVQZ calculations performed on the [ArnH]+ clusters”

The method and the basis set should be consistent for geometry optimization. In addition, geometry optimizations do not require very large basis sets. For example, cc-pVQZ is enough.

2. The reference for the Dunning basis set is missing: [1] Mol. Phys. 96, 529-547 (1999) [2] J. Mol. Struc-THEOCHEM 388, 339-349 (1996)

3. Because the systems studied in this work are in the gas phase, can the authors provide a brief discussion on the temperature effect?

4. page 13, Figure 11

Three magic numbers are shown in this figure, are these numbers related to the symmetry of the corresponding structure?

5. Can the designed machine learning model be used for a much larger cluster size? Can the authors comments on the extrapolation capabilities of this ML model?

Comments on the Quality of English Language

No major language or grammar problem found.

Author Response

Referee's Comments and Authors' reply

In this work, the authors investigate the structural arrangements and energetic stabilities of small protonated argon clusters using first-principle initio computations. It shows four-body contributions are crucial for accurately describing potential surfaces in clusters with more than three Ar atoms. Analyzing structural stability using these models and evolutionary programming, the study identifies energetically favored ArnH+ structures at magic sizes n=7, 13, and 19, forming Ar-pentagon rings perpendicular to the ArH+Ar core. These findings align with ion yield data from mass spectrometry and demonstrate the model's effectiveness.

This work can be interesting for the computational chemistry and the industrial community. I would like to ask the authors to consider the minor comments below.

Authors' reply: We thank the reviewer for the supportive discussion on our work, and we have addressed all points raised in the revised ms.

1. page 2, line 88

Figure 1 and Table S1 (see supplementary material) present our results on optimized structures and binding energies derived from the MP2/AVQZ and CCSD(T)/AV6Z or CCSD(T)-F12/AVQZ calculations performed on the [ArnH]+ clusters”

The method and the basis set should be consistent for geometry optimization. In addition, geometry optimizations do not require very large basis sets. For example, cc-pVQZ is enough.

Authors' reply: We agree with the referee's comment. For this reason, we present results for different methods and basis sets in Table S1.

2. The reference for the Dunning basis set is missing: [1] Mol. Phys. 96, 529-547 (1999) [2] J. Mol. Struc-THEOCHEM 388, 339-349 (1996)

Authors' reply: Following the referee's suggestion, we have now included 4 new references in the bibliography regarding the basis sets used in this work (see refs. 34-37 in the revised ms).

3. Because the systems studied in this work are in the gas phase, can the authors provide a brief discussion on the temperature effect?

Authors' reply: Depending the value of the temperature such effects are expected to influence mainly larger size clusters, as the number of the low-lying isomers is increasing. At low temperature (few K) and for the smaller size clusters we expect broader radial and angular distributions, while as temperature and cluster size increase such effects may become significant and should be investigated properly. Thus, we have added a note on page 14, line 348 in the revised ms.

4. page 13, Figure 11

Three magic numbers are shown in this figure, are these numbers related to the symmetry of the corresponding structure?

Authors' reply: The numbers shown in Figure 11, as mentioned in the text correspond to the number n of Ar atoms in the corresponding cluster.

5. Can the designed machine learning model be used for a much larger cluster size? Can the authors comments on the extrapolation capabilities of this ML model?

Authors' reply: The present ML model has employed in the present work and has shown very good agreement with experimental ion yield measurements for ArnH+ clusters with n up to 25. As we mentioned in the last section we plan to investigate in the near feature the influence of quantum effects at low temperatures, as they expecting to affect the assignment of patterns observed experimentally for larger clusters up to 50 Ar atoms.